# Medication Reconciliation as Part of Admission Management—A Survey to Improve Drug Therapy Safety in a Urology Department

**DOI:** 10.3390/pharmacy12040122

**Published:** 2024-08-06

**Authors:** Yvonne Remane, Luisa Pfeiffer, Leonie Schuhmann, Annett Huke, Jens-Uwe Stolzenburg, Thilo Bertsche

**Affiliations:** 1Pharmacy, Leipzig University Medical Center and Medical Faculty, Liebigstrasse 20, 04103 Leipzig, Germany; yvonne.remane@medizin.uni-leipzig.de (Y.R.); luisa.pfeiffer1@gmx.de (L.P.);; 2Drug Safety Center, Leipzig University and Leipzig University Medical Center, 04103 Leipzig, Germany; 3Clinical Pharmacy, Institute of Pharmacy, Medical Faculty, Leipzig University, Brüderstrasse 32, 04103 Leipzig, Germany; 4Department of Urology, Leipzig University Medical Center, Liebigstrasse 21, 04103 Leipzig, Germany; annett.huke@medizin.uni-leipzig.de (A.H.);

**Keywords:** medication reconciliation, medication discrepancies, pharmacist, hospital admission

## Abstract

Complete medication reconciliation during hospital admission is the rationale for further treatment decisions. A consecutive, controlled intervention study was conducted to assess discrepancies in medication reconciliation performed by nurses of the Urology Department compared to the Best Possible Medication History (BPMH) established by pharmacists. This study included pre-intervention (control group, CG), nursing training as a pharmaceutical intervention, and post-intervention (intervention group, IG) groups. The discrepancies were classified as “Missing” (not recorded but taken), “Added” (additionally recorded) “Strength” (incorrect documented dosage), “Intake” (incorrect intake time/schedule), “Double” (double prescription), and “Others” (no clear assignment). Additionally, high-risk drug subgroup discrepancies were particularly prevalent and were evaluated. Training success was compared concerning discrepancies in the CG and IG. Generally, the percentage of discrepancies per patient found was lower in the IG than in the CG (78.1% vs. 87.5%, significantly). The category most identified was “Missing” (IG, 33.3% vs. CG, 35.2%). Overall, a discrepancy of 7.4% each (discrepancies: IG, 27 vs. CG, 38) was determined for high-risk drugs while “Missing” occurred (77.8% vs. 52.6%, out of 7.4%). Despite nursing training only partially reducing discrepancies, the implementation of medication reconciliation using BPMH by pharmacists could improve the process, especially for high-risk drugs.

## 1. Introduction

Continuous improvement in drug therapy safety results from intensive interdisciplinary collaboration. The focus is mainly on process optimization, which includes the following pharmaceutical activities: creation of a medication plan, medication reconciliation, and medication management. However, these have not yet been implemented nationwide. Medication reconciliation, in particular, was a largely unknown process step in German hospitals until a few years ago. Therefore, the WHO’s initiative, “Action on Patient Safety—High 5s” project, focused on implementing medication reconciliation, including the correct transmission of medication at care interfaces and good communication between healthcare professionals, including pharmacists [1]. The incomplete transmission of medication data and an insufficient medication list is regarded as one of the most significant potential risks when admitting patients to the hospital. Intentional or unintentional discrepancies frequently occur between outpatient and inpatient medication [2]. In contrast to unintentional discrepancies, intentional changes were defined as made knowingly, such as adding, changing, or discontinuing medications [2]. In this context, medication reconciliation has proven to be an effective method to clarify discrepancies and identify and reduce risks at care interfaces [2,3,4,5]. Medication reconciliation should provide a complete medication history to optimize therapy afterward effectively. Information from the patient’s chart and files, including current laboratory findings, previous physician’s letters, the Federal Medication List, and patient interviews, could be used to obtain a complete medication history. The comprehensive Best Possible Medication History (BPMH), i.e., the best possible determination of previous drug therapy, is the basis for this purpose [6]. The BPMH is a systematic recording of current medication and should be the central part of medication reconciliation. It differs from usual medication recording through its scope and systematic structure, such as using a standardized questionnaire [6]. Additionally, various assessment tools are available to assess the clinical relevance of the identified discrepancies [7,8,9,10]. As part of the WHO’s High 5s concepts, a specific list of high-risk drugs has been introduced and validated by Doerper et al. [11]. In routine care, nursing staff frequently record outpatient medication. Medication reconciliation is a critical process for admission to the hospital, and it has only been handed over to three very responsible nurses in the Urology Department. However, as we have already identified discrepancies in this small group of people, we wanted to strengthen their expertise and focus through targeted training. Therefore, the results of medication reconciliation by nurses were compared to BPMHs by pharmacists before and after offering a training program to the nursing staff.

## 2. Materials and Methods

### 2.1. Setting and Study Design

After receiving ethics approval from the responsible local ethics committee, this study was performed between May and October 2023. This study was conducted in the Urology Department and included three study periods: 1. a six-week pre-intervention period (for the CG) to examine the routine process; 2. a Pharmaceutical Intervention Nursing Staff Training period over nine weeks to implement process optimizations such as increasing awareness of potential pharmaceutical issues, conducting training sessions, and introducing a checklist to assist with medication reconciliation for nursing staff (additionally, patients were informed to bring their medication lists with them at the time of admission, enhancing the efficiency of the process); and 3. a six-week post-intervention period (for the IG) to compare discrepancies between the IG and CG. Both periods for the IG and CG happened outside the summer holidays. The recruitment numbers of at least 200 patients in the CG and IG were targeted. Based on the results of an independent pilot evaluation, we predicted that 90% of patients in the CG and 80% in the IG would have at least one discrepancy (primary endpoint defined as discrepancies in medication reconciliation performed by nurses in the Urology Department with the Best Possible Medication History performed by pharmacists). Assuming rates in this range, a two-sided Chi-square test with a significance level of a = 0.05 and a sample size of at least 199 per group would yield a power of 1 − b = 0.80 for the primary endpoint (patients with at least one discrepancy). The inclusion criteria were elective admission for an inpatient stay, written informed consent by the patient, and sufficient German-speaking ability to participate. As is routine, three nurses with working experience of more than ten years were involved in the medication reconciliation process for elective patients. They participated fully in both the IG and CG.

### 2.2. Pre-Intervention Period (for the CG)

At our Urology Department, the nursing staff used an electronic system for documentation (ID Medics) to carry out medication reconciliation. After the nurses had performed medication reconciliation, pharmacists interviewed the patients again using the BPMH methodology. Recorded medications were checked for discrepancies.

### 2.3. Nursing Training as Pharmaceutical Intervention

After the six-week pre-intervention period, the nursing staff were invited to participate in training. Based on an assessment of the discrepancies in the CG, a training concept was designed and implemented in several steps. In the first step, each of the three nurses observed a patient individual pharmacist medication interview over two hours. The aim was to learn from the pharmacist how to ask patient-specific questions to obtain a complete medication record. In the second step, individual training sessions were held in small groups, including the three primary carers. As all nursing staff carry out medication reconciliation for non-elective patients, they could also participate in this training. The training conducted by the pharmacist comprised a theoretical part of 45 min. In this session, exactly how a BPMH should be performed was explained. The following points focused on ensuring that the medication plan was current and comprehensive, as well as presenting suitable questioning techniques and the most common errors with illustrative examples. A checklist was presented to support the process. In this step, a total of 21 nurses participated. Training was offered four times, with 2 to 10 nurses and 1 to 4 trainers participating, and a pharmacist carried out the training. The content was specifically tailored to identify problems, high clinical risks, or particular patient-specific needs (Table 1).

In the third step, a practical exercise was offered. Identified problems, such as recording dosage schedules or pausing medication administration, were recorded. In this process, IT-supported practical examples of the ID Medics program were used. As a last step, the nursing staff were given the r checklist as a handout for the medication reconciliation in the IG.

### 2.4. Post-Intervention Period (for the IG)

The nursing staff training was followed by a six-week post-intervention period. In addition, the patients were asked to bring all documents related to their existing medication therapy during hospital admission—the BPMH. The documentation of discrepancies and patient interviews were conducted analogously to the CG in the previous pre-intervention period. The data were analyzed, compared, and documented concerning their discrepancies, as in the CG. No additional tools were used.

### 2.5. Parameters

All admitted patients in these study periods who met the inclusion criteria were consecutively enrolled. All patient data were entered into a database in an anonymized, aggregated form for further data processing. In addition to complete medication history, the following data were recorded: age, sex, weight, size, admission diagnosis according to ICD 10, secondary diagnoses (if these were recorded), interview duration, and admission day. Then, the data were digitally recorded according to the BPMH guidelines. Following patient interviews, medication reconciliation established by nursing staff was compared to the BPMH performed by a pharmacist, and discrepancies were checked.

The following parameters were recorded for all documented discrepancies: name of the drug concerned, strength and dosage for the form of drug, type of discrepancy, and classification of drug discrepancies as defined according to the WHO’s template. The medication reconciliation process was analyzed as a retroactive model, as follows: 1. routine process (recording outpatient medication by nursing), 2. BPMH (carried out by clinical pharmacists), 3. Comparison of discrepancies between routinely recorded medication and the one from the BPMH process, and 4. Clarification of discrepancies. The BPMH was performed similar to the guideline “Best possible medication history”, based on the 2012 published High 5s SOPs “Medication Reconciliation” of the German “Aktionsbuendnis Patientensicherheit”, a quality assurance concept standardized by the Medical Center for Quality in Medicine, a joint institution of the German Medical Association and the National Association of Statutory Health Insurance Physicians [12]. All discrepancies identified were discussed and clarified with senior clinical pharmacists and then with the ward physician; then, these were evaluated.

### 2.6. Discrepancies

All differences that occurred between pharmaceutical and nursing staff were classified as discrepancies. All discrepancies were recorded, even if several occurred per patient and/or medication. In addition, dietary supplements, vitamins, minerals, and herbal preparations were also recorded, as they might have influenced drug therapy or the anesthesia of a surgical procedure. Therefore, all identified discrepancies were adapted to Schmitz et al. and classified into the following subcategories: “Missing”, “Added”, “Strength”, “Intake”, “Double”, and “Others” [13]. If drugs were not recorded, they were classified as “Missing”. Additionally, documented drugs were categorized as “added”. For “Intake”, the incorrect time of drug administration described the change in administration time only, which did not result in a change in the total daily dose. Multiple drug prescriptions were counted as “Double”. All other discrepancies in the prescription process were summarized under “Others”.

### 2.7. Statistical Analysis

A descriptive analysis of the collected data was carried out using Microsoft Excel (Microsoft Corporation, Redmond, WA, NY, USA). In addition to Microsoft Excel, SPSS (V29.0, IBM) was used for graphical representation. All continuous variables were described as median with an appropriate interquartile range. Categorical data were presented as absolute and relative frequencies (percentages). The statistical analysis was performed using SPSS, where the Chi-square test was calculated. The results of the tests are reported as two-tailed *p*-values. A *p*-value ≤ 0.05 was considered statistically significant.

## 3. Results

### 3.1. Characteristics of Participating Patients

Two hundred patients taking 1277 drugs were enrolled in the CG; 49 (24.5%) were female, and 151 were male (75.5%). The median age of the patients was 60 years (spread over 24–92 years; Q25: 53 years; Q75: 67 years) with a median outpatient medication of 6 (Q25: n = 3; Q75: n = 9). In the IG, a total of 178 patients using 979 medicines (not registered in the control group) were enrolled; 33 patients (18.5%) were female, and 145 were male (81.5%). Their median age was 67 years (spread over 20–93 years; Q25: 59.5 years; Q75: 74.5 years) with a median of five drugs in outpatient medication (Q25: n = 2.75; Q75: n = 8). The characteristics of all enrolled patients are shown in Table 2.

### 3.2. Discrepancies in Intervention and Control Group

All 3 primarily responsible nurses for medication reconciliation (100%) and 18 of the 34 additional nurses (52.9%) participated in the training program. A median of two discrepancies per patient (Q25; Q75 = 1; 3) was documented in both groups (IG and CG). The BPMH recorded at least one discrepancy in 87.5% (n = 175 out of 200) of the CG patients, who had at least one discrepancy, while in 12.5% (n = 25) of the study participants, no discrepancies were found.

In comparison, 78.1% (n = 139 out of 178) of the study participants in the IG had at least one discrepancy recorded by the BPMH; in 21.9% (n = 39), no discrepancies were found. The maximum number of identified discrepancies per patient was 10 (0.6%, n = 1) in the IG vs. 12 (0.5%, n = 1) in the CG. Discrepancies of all taken drugs were IG—37.3% vs. CG—40.5%. In general, the percentage of at least one discrepancy was found to be significantly lower in the IG than in the CG (78.1% [n = 139/178] vs. 87.5% [n = 175 out of 200], Chi-square *p* = 0.015). In the IG, in the category “Missing”, we observed only a slight reduction (33.3% [n = 326 out of 979 drugs; IG] vs. 35.2% [n = 449 out of 1277 drugs] in the CG). In the group of OTC preparations, in the comparison between the CG and IG, the discrepancies in the “Missing” group decreased from 42.0% (n = 217) to 37.3% (n = 136). The PRN medication group recorded a decrease from 49.3% in the CG (n = 255) to 39.5% in the IG (n = 144). Also, a reduction was documented in the category of modified dosage forms (IG, 22.5%, number of discrepancies n = 82 vs. CG, 19.7%, n = 102), whereas in the high-risk drug group (IG, 5.8%, n = 21 vs. CG, 3.9%, n = 20), an increase in discrepancies in the category “Missing” was identified (Figure 1).

A total of 7.4% (IG n = 27; CG n = 38 discrepancies) refer to high-risk medications. Most of these belong to the category “Missing”, i.e., undocumented but having taken medicines—in the IG, 5.75% (n = 21 out of 365) vs. in the CG, 3.87% (n = 20 out of 517)—followed by incorrect intake time/incorrect intake schedule, documented as “Intake”, in the IG, 0.27% (n = 1) vs. in the CG, 2.51% (n = 13); “Strength” (incorrect documented dosage), in the IG, 1.1% (n = 4) vs. 1.55% (n = 8); and “Added” (additional medication recorded), in the IG, 0.27% (n = 1) vs. 0.38% (n = 2). The category “Other”, which includes discrepancies that could not be assigned, has a share in the IG at 0% vs. in the CG at 0.19% (n = 1). Most discrepancies in the category “Missing” in the IG at 5.8% were observed in the classification of high-risk drugs (n = 21), specifically beta-blockers, with a rate of 2.2% (n = 8), followed by anticoagulants and antitumor agents at 1.1% each (n = 4). In the CG, the number was 3.9% (n = 20); however, discrepancies were found for the following high-risk drugs: antibiotics at 1.93% (n = 10), followed by beta-blockers at 0.6% (n = 3), and antitumor agents at 1.0% (n = 5). Furthermore, discrepancies were found in the drug groups of insulins, immunosuppressants, heparins, and anticoagulants (Table 3).

Despite the implementation of training and procedural changes, including the introduction of a checklist for nursing staff and patient notification of the need to bring their medication list on admission, the persistently high prevalence of discrepancies between nursing-staff-recorded outpatient medication and the pharmaceutical BPMH indicates the necessity of establishing a pharmaceutically supported medication reconciliation process.

## 4. Discussion

### 4.1. General Considerations

In the Urology Department, predominately, patients at risk due to age and polymedication are treated. It can, therefore, be assumed that in this patient group, there are often unintended discrepancies between outpatient and inpatient prescriptions to be expected. In our study, most patients were male and were admitted for malignant neoplasms of the prostate according to ICD C61. The discrepancies between routine medication reconciliation by nursing were compared to the BPMH by pharmacists. In our CG, there were more than 40% discrepancies in all the taken drugs identified. So, we examined whether structured training in small groups for the nursing staff influences the number of discrepancies in medication reconciliation. It was shown that training led to a significant reduction in discrepancies per patient (IG, 78.1% vs. CG, 87.5%, Chi-square *p* = 0.015) and in the discrepancies concerning all taken drugs (IG, 37.3% vs. CG, 40.5%), although they all remained at a high level. This crucial process step observed time intensity and demanding workload associated with incomplete or non-existent medication lists. Factors that have been determined to influence nursing medication reconciliation have already been described in the literature; in particular, demanding working environments, high workloads, distractions, and frequent interruptions increase the risk of medication errors [14]. Regarding patient safety, the nursing staff’s low training success is unacceptable due to the urological patients’ polymedication, age, and associated risks, as is known from the literature. These patient groups will benefit from pharmacists’ medication reconciliation [15].

### 4.2. Best Possible Medication History

The Best Possible Medication History method was used to compare the routine process of reconciling medication by nursing staff. The implementation of the BPMH as part of medication reconciliation is a very time-consuming process. Obtaining all the necessary information from different sources requires much personal effort. Most are still paper-based or only available after consultation with patients and/or attending physicians. Some studies showed reduced time spent using an electronic patient record [16,17,18,19]. According to the WHO’s High 5s project, several sources for the BPMH were included [17,18], and a structured guideline covering both food supplements and drugs with modified dosage release was implemented. In 2010, a US study found that 66% of patients took vitamins, 35% took OTC preparations, and 21% took dietary supplements [20]. Therefore, it can be assumed that these preparations should also be attended to. During our patient interviews and on specific request in accordance with BPMH guidelines, many patients only reported the intake of their dietary supplements and OTC preparations when specifically asked to and gave memorable examples. In our observation, patients do not report these intakes on their own initiative in routine drug surveys, and therefore, they are often not recorded in prior charts. This could be confirmed by categorizing our discrepancies. The literature shows that discrepancies are usually frequently associated with dietary supplements (i.e., vitamins and minerals) and drugs in modified dosage forms. Dietary supplements, in particular, have a high potential for interactions with current drug therapy; e.g., supplement use may also mask symptoms, making it more challenging to identify and treat the underlying disease [20,21].

### 4.3. Discrepancies

In the CG, 87.5% of patients had at least one documented discrepancy, while in the IG, the corresponding number was 78.1%. The percentage of discrepancies per patient was significantly lower in the IG than in the CG (Chi-square *p* = 0.015). Compared to the literature, the collected amount appears to be above the mean level, as in many studies, 8% to 70% of patients were affected by at least one discrepancy [4,8,10]. The different definitions of discrepancies and focusing on certain drug groups led to high heterogeneity within the studies [15,22]. Studies similar to our own, observing medication and documentation errors, also showed that 93.6% to 97% of patients had at least one discrepancy [23,24]. We identified a median of two discrepancies per patient in both the IG and CG. Compared to the literature (1.2 and 2.6 per patient), our results are still in the upper range [17,25,26]. One reason may be that different definitions of the subgroups and categories directly influence the results. Additionally, it is already known that the amount of medication taken is proportional to the occurrence of medication errors [27]. In our study, the median number of medications patients took was 5 to 6. Gleason et al. showed a risk increase with an odds ratio of 1.21 [27]. Remarkably, the “Missing” category is primarily dominant; 25% to 79.6% of the identified discrepancies in the literature can be subsumed under this category [18,23].

### 4.4. Clinical Relevance

The success of the training was assessed by comparing nursing medication reconciliation with the BPMH method. When comparing the discrepancies in the high-risk drug group, they occurred in the post-interventional period in 7.4% (n = 27) and in the pre-interventional stage in 7.4% (n = 38). Unfortunately, regarding high-risk drugs, the supposed drug safety risks for patients remained unchanged after the training. Most of these discrepancies were assigned to the “Missing” category, i.e., the patient’s drug therapy was incompletely recorded. The unintentional discontinuation of medication, e.g., before surgical procedures, can pose a potential risk for the patient. Inadequate treatment for diabetes mellitus leads to an imbalance in blood sugar levels, which can lead to the postponement of a surgical intervention [28].

In particular, phytopharmaceuticals such as Gingko biloba, Hawthorn, milk thistle, etc., and nutritional supplements have an increased risk of bleeding. Therefore, these preparations are contraindicated before a surgical procedure and should be stopped for up to two weeks before intervention [29]. Despite reducing discrepancies in the OTC preparation group from 42% in the pre-interventional period (n = 217) to 38.4% (n = 140) in the post-interventional stage, OTC preparations are often not recorded in the routine process after training. The reason could be that patients frequently do not perceive these medications as drug therapy (e.g., phytopharmaceuticals and over-the-counter analgesics). These medication reconciliation discrepancies are clinically relevant and associated with a worse outcome.

### 4.5. Implementation of Medication Reconciliation

The high incidence of discrepancies between drug collection by the nursing staff and the pharmaceutical BPMH confirms that establishing a medication reconciliation process is urgently needed. Medication reconciliation performed by pharmacists has already been implemented in centralized patient admission management but not in the Urology Department. A safe medication admission process requires a high level of pharmacological expertise and is associated with much time. Pharmacists can best provide focus on the assessment and management of drug-related problems in this highly critical process; e.g., Pevnick et al. showed that pharmacist-performed medication reconciliation could reduce medication errors by more than 80% [15].

### 4.6. Limitations

This study has limitations that should be considered when interpreting the results. Firstly, it can be assumed that the BPMH, due to the heterogeneity of available data sources, could not always correctly record the patient’s current drug therapy and was based on the systematic interview provided by the patients. A brown bag analysis was not feasible in the inpatient setting. Secondly, patients may not remember their previous medications when asked about them. It is also possible that history taking may be incomplete if not all information is remembered (recall bias). Thirdly, the urological patient group was particular. Uneven gender distribution and the influence of specific risk factors could be observed. The female gender is one of the risk factors for drug-related problems [30]; therefore, the potential risk of discrepancies in this study’s population was reduced. Thus, the results are not necessarily transferable to other specialist areas or clinics. Fourthly, the success of the training is dependent on the personal attitude and on the level of preexisting knowledge of the primary nurses.

## 5. Conclusions

Our study aimed to improve drug safety and analyze the medication reconciliation process in the Urology Department. Despite the training of the nursing staff, there were still discrepancies in patients’ admission medication, both pre-and post-intervention, particularly for high-risk drugs. It was shown that the discrepancies per patient statistically decreased after the training, but the training success remained low. Especially from the unchanged proportion of discrepancies relating to high-risk drugs, there are still patient risks left. Medication reconciliation should be the best practice for every patient on admission to the hospital. As nurse training did not achieve the expected success, further research is required to identify whether alternative training strategies could enhance the success rate or whether pharmacist-supported medication reconciliation could bridge the gap to complete the medication records of the patients.

## Figures and Tables

**Figure 1 pharmacy-12-00122-f001:**
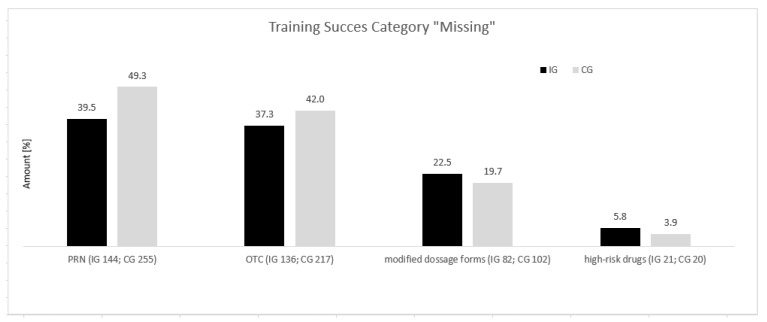
Comparison of training success in the category “Missing”, i.e., undocumented but administered medication in the CG (n = 517) vs. IG (n = 365).

**Table 1 pharmacy-12-00122-t001:** Development of the training program.

Structure of the Training Program:	Examples
Ensuring that the medication the plan is current and comprehensive	Inquiry about treatment adherence/changes
	Inquiry of missing information on dosage form, time of administration, strength, dosage regimen
	Inquiry about modified forms, PRN, OTC
Presentation of suitable questioning techniques	Open-question technique, avoidance of leading questions
	Giving examples
	Checking the number of tablets
Presentation of most common errors with illustrative examples	Incorrect dosage schedules
	Double medication
	Outdated medication plan
Practical exercises for electronic recording of modified dosing regimens	Durogesic SMAT^®^ 50 µg/h patch with the dosing schedule: 1 patch every three days, last changed yesterday
	Madopar^®^ 125 mg tablets with dosing regimen: 06:30 0.5 tablet, 09:30 one tablet, 12:00 0.5 tablet, 14:30 one tablet, 17:00 0.5 tablet
	Eliquis^®^ 5 mg film-coated tablets with the dosing regimen: 1-0-1 tablet, paused since the day before yesterday

**Table 2 pharmacy-12-00122-t002:** Characteristics of enrolled patients in the pre- and post-intervention period; admission diagnosis using the following ICD-10 categories: C (malignant neoplasm; C61 prostate, C64 kidney, C67.8 urinary bladder), D (in situ neoplasms, benign neoplasms, and neoplasms of uncertain and unknown assigned behavior), and N (diseases of the genitourinary system).

		Control Group			Intervention Group		
		Median	(Q25; Q75)	Spread	Median	(Q25; Q75)	Spread
Age [y]		60	(53; 57)	24–92	67	(59.5; 74.5)	20–93
Medication		6	(3; 9)	0-22	5	(2.75; 8)	0–21
Sex	**Female**	49 (24.5%)			33 (18.5%)		
	**Male**	151 (75.5%)			145 (81.5%)		
Admission Diagnosis	**C Category Neoplasm** (malignant)	**117 (58.5%)**			**85 (47.8%)**		
	C 61 (prostate)	52 (26%)			45 (25.3)		
	C 64 (kidney)	19 (16%)			11 (6.2%)		
	C 67.8 (urinary bladder)	32 (9.5%)			17 (9.6%)		
(ICD-10)	**D Category Neoplasm**	**5 (2.5%)**			**8 (4.5%)**		
	in situ/benign, uncertain/unknown						
	**N Category**	**71 (35.5%)**			**75 (42.1%)**		
	diseases of the genitourinary system						
	**Others**	**7 (3.5%)**			**4 (2.2%)**		

**Table 3 pharmacy-12-00122-t003:** Categories of recorded drug groups and high-risk drugs of medication reconciliation according to Doerper et al., control group (CG) vs. intervention group (IG).

	Missing [%] *		Strength [%] *		Intake [%] *		Added [%] *		Double [%] *		Others [%] *	
	IG	CG	IG	CG	IG	CG	IG	CG	IG	CG	IG	CG
PRN ^1^ (n = 149/257)	39.5	49.3	−	−	0.6	−	−	−	−	−	0.8	0.4
OTC (n = 140/217)	37.3	42.0	−	−	0.6	−	−	−	−	−	0.6	−
Modified Forms (n = 87/107)	22.5	19.7	0.3	0.4	0.3	0.4	0.6	0.2	−	−	0.3	−
High-Risk Drugs (n = 27/38)	5.8	3.9	1.1	1.6	0.3	2.5	0.3	0.4	−	−	−	0.2
*Antibiotics*	*0.6*	*1.9*	*−*	*−*	*−*	*−*	*−*	*0.2*	*−*	*−*	*−*	*0.2*
*Beta blocker*	*2.2*	*0.6*	*0.6*	*0.6*	*−*	*0.8*	*−*	*−*	*−*	*−*	*−*	*−*
*Insulin*	*−*	*−*	*0.3*	*0.4*	*−*	*0.6*	*−*	*−*	*−*	*−*	*−*	*−*
*Antitumor drugs*	*1.1*	*1.0*	*−*	*−*	*−*	*−*	*−*	*−*	*−*	*−*	*−*	*−*
*Immunosuppressant*	*0.3*	*−*	*−*	*0.4*	*−*	*0.8*	*−*	*−*	*−*	*−*	*−*	*0.2*
*Heparine*	*0.3*	*0.2*	*−*	*0.2*	*−*	*−*	*−*	*0.2*	*−*	*−*	*−*	*−*
*Anticoagulants*	*1.7*	*0.2*	*0.3*	*−*	*−*	*0.4*	*0.3*	*−*	*−*	*−*	*−*	*−*

^1^ PRN = medication, if required. “Missing”—not recorded but administered medication; “Strength”—incorrect documented dosage; “Intake”—incorrect intake time/incorrect intake schedule; “Added”—additional recorded medication; “Double”—double prescription; “Others”—No clear assignment. * Total discrepancies: CG = 517; IG = 365.

## Data Availability

The data can be requested from the corresponding author upon reasonable request.

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
