# Peer review of "Medication Reconciliation as Part of Admission Management—A Survey to Improve Drug Therapy Safety in a Urology Department"

_pharmacy, 2024, doi:10.3390/pharmacy12040122_

Round 1
Reviewer 1 Report
Comments and Suggestions for Authors
The authors' study, which aims to improve drug therapy safety through training nursing staff in medication reconciliation, is a commendable endeavor. While the results are likely to be of interest to the readership of Pharmacy, I feel compelled to raise several concerns that require the authors' attention (please see below). Additionally—it is unusual for me to directly point this out—I must mention that this manuscript has a notable level of inconsistency and occasional inaccuracies in presenting terms and results, as well as instances of misplaced punctuation and a few grammatical errors, which is dreadfully ironic considering the focus of the paper (errors in medication reconciliation). These issues collectively detract from the clarity of your findings. In my view, many of these issues could have been identified and addressed by the authors before the manuscript was submitted for publication.
Major comments:
1. Your study (performed between May and October) includes pre-intervention and post-intervention periods. Any differences between these two periods could, of course, be attributed to the study intervention (nursing training), but may have been influenced by a period effect (e.g., nursing staff leaving for vacation, which leaves the remaining staff with relatively more workload, which raises the likelihood of errors being made). Please discuss this study limitation and how this could have influenced your findings.
2. The authors describe that “the BPMH was performed as a retroactive model after the medication reconciliation considering the following aspects: routine process, outpatient medication, BPMH, comparing for discrepancies, clarification of discrepancies.” This is a form of circular definition or self-referential definition, where the BPMH is defined in terms of itself or in terms of components that include itself, creating a loop in the definition. Please clearly describe how exactly the “true” medication usage of patients, which is the touchstone of your study, was established.
3. Please describe what proportion of the nursing staff participated in training. Then, please clarify whether medication reconciliation was performed by a mix of trained and untrained nurses or strictly trained nurses.
Minor comments:
1. The word “an” can be omitted from the title. The phrase "admission management" is used as a compound noun, where "admission" functions as an attributive noun describing "management." In such cases, it's common to omit articles like "a" or "an" before the noun that follows. This omission helps maintain clarity and conciseness in the title.
2. The Abstract states that “[t]he results of phase one and three were compared in terms of training success.” It is not clear what phase one and three are referring to here, as they haven’t been defined earlier in the Abstract. Perhaps the authors are referring to pre-intervention and post-intervention; if so, I recommend the authors to use these terms then.
3. The Abstract states that “a discrepancy of each 7.4% (n=27/38) was identified”. It is unclear to me what the proportion “27/38” refers to, as this would yield 71%. Please clarify; same comment for Section 3.2.
4. The Abstract was somewhat difficult to understand due to technical shortcomings, including use of undefined terminology and a conclusion that partly touches upon a pharmacist-related aspect, while no pharmacist-related results were presented; instead, the Results subsection exclusively pertains to nursing training as a pharmaceutical intervention. In addition, there is some misplacement of punctuation marks, non-standard use of first-person phrases ("we performed," "we identified X" rather than "a study was performed," "X was demonstrated"), and inconsistent capitalization of terms. Careful rewriting is advised to clarify your findings.
5. The Introduction section states that “[t]he greatest potential risk when admitting patients to hospital lies in the complete transmission of medication data”. This suggests that the risk lies in the data not being fully or accurately transmitted, which could lead to potential errors or issues during patient admission. Therefore, in the context of the sentence, "incomplete" would be more appropriate.
6. The manuscript contains several grammatical and punctuational errors similar to those previously pointed out. Moving forward, I will refrain from pointing out additional instances. However, I advise that you consider having the manuscript reviewed by an experienced English-speaking colleague or using a reputable language-editing service while revising your manuscript.
7. The authors mention that the parameter “gender” was recorded. In medical context, “sex” is the preferred term for biological forms, whereas “gender” is limited to its meanings involving behavioral, cultural, and psychological traits.
8. The authors describe that “[a] p-value ≥0.05 was considered statistically significant.” I assume a p-value <0.05 is meant here.
9. The authors seem to refer to the interquartile range (IQR) as “Q25/Q75”. I am not familiar with “Q25/Q75” and to me the solidus (or slash “/” symbol) implies this is the ratio between the 25th percentile and the 75th percentile. Please clarify. In addition, providing the actual values of the 25th and 75th percentiles, rather than the difference between the 25th and 75th percentiles, is more helpful to the reader, because the difference itself is unanchored.
10. In Table 1, “Q25/Q75” seems to be called interdecile range (IDR) here. Thus, either “Q25/Q75” or “IDR” is incorrectly used in the manuscript. Please correct and choose one term throughout.
Comments on the Quality of English LanguageSee above.
Reviewer 2 Report
Comments and Suggestions for Authors
I read with interest your project that detailed an educational initiative to impact the rate of medication documentation errors. The paper appropriately used prior literature to form some methodologies related to classification of errors found which is helpful for external validity of the work. Below is a summary of areas needing attention.
English grammar and word choice will require attention as there are multiple areas in the manuscript which are unclear. For example, "complete transmission" at the bottom of the first page would usually imply electronic transmission of data, but within the context it appears it means a complete and accurate medication list. The next sentence indicates intentional and unintentional discrepancies. I am not what what an intentional discrepancy would be (is this really a discrepancy if it was intended to be left off the list or there was a different dose intended?).
Methods
It is not clear why 200 patients were selected. Was this based on a power analysis? What was the rationale for that number?
I liked the three stage approach with a baseline, intervention, and post intervention stage. However, I have concerns for bias and incomplete description of the methods. For example, there is not a mention of how many nurses were involved (before, training, and after), there was not any mention of why it took 9 weeks for training, what the sessions included, how many sessions there were, the number of trainers, etc... All of these factors could influence the results. Additionally, the BPMH needs further description as to what this process looked like. This is also a limitation since there will be recall bias on the part of the patient and they may remember medications not collected by the nurse (e.g., I forgot to tell the nurse about this when they came in previously). The methods also state nursing staff were "invited" to participate. If this was not mandatory, then these data (% who completed) need to be reported. In general, the training needs a lot more description of what happened including use of relevant examples (see theoretical example section of methods). There was mention of a checklist for medication reconciliation. Was this included as part of the training or simply provided to nurses as the last stage?
The post-intervention paragraph mentions a "survey". This implies perception and recall of an event, rather than what you did which was compare to BPMH. There was also mention that patients were informed to bring medication documents to the hospital for this stage. Was this not done in the baseline stage? If there was a difference, this is another significant confounder for your data. The last sentence in this paragraph is not clear.
Parameter paragraph lists BPMH twice in the same sentence. Additionally, there are terms which are unclear/undefined including "routine process" and "comparing" or "clarifying" discrepancies. As written, this project could not be reproduced because of lack of operational definitions or adequate descriptions of the methodology.
Further on this concept is how discrepancies were actually handled. For example, what if a dose and frequency was incorrect for a single drug? Was this counted as 2 or 1?
Statistical analysis. There was inconsistent use of median, mean, average, IDR and IQR which did not follow the methodology.
Results section was generally very difficult to read because of the way the data are presented (Q25/Q75) rather than IQR. There was also inconsistent use of terms (outpatient medication vs. primary care).
Table 1 could include the results of the statistical tests (presumed not-significant) and a listing of the ICD codes as descriptive text within the table (there is real-estate for this). It would also be helpful to list the number of nursing personnel before and after and perhaps some demographics about their years of experience as a nurse. If all did not complete the training, then a percentage of those that did could be helpful.
Section 3.2. These data would be better to put into a table. Additionally, paragraph 1 and 2 are both the same thought to splitting them out makes it more confusing to read.
Table 2 contains terms that may not be known in other parts of the world including "On-demand" and "Special Forms". These should be defined. There could be p-values added to the table, where appropriate.
The bottom of page 5 lists "resp. 3.9%, n=20 vs. 5.8%, n=21). It is not clear what these results are.
There was data included in the discussion which were not in the results (that I could find). For example, 40% of discrepancies in the CG.
The discussion section needs to include a discussion of interventions (e.g., educational) and a comparison to the current results. In essence, this did not work so what are the next steps?
Comments on the Quality of English LanguageSee text above.
Round 2
Reviewer 1 Report
Comments and Suggestions for Authors
Thank you to the authors for addressing the comments I provided. I have no further remarks.
Comments on the Quality of English Language-
Author Response
Thank you very much for taking the time to review this manuscript.
Reviewer 2 Report
Comments and Suggestions for Authors
The paper is much improved. The low training of typical nursing staff was still low which is a major contributor and should be listed as a limitation. Although pharmacists were the source this was compared with, the conclusion should focus on the study design. For example, why does this not point to a need for better training with nurses? Barriers to effective medication reconciliation were not explored which could be another limitation.
Comments on the Quality of English LanguageImproved but still areas that could be optimized.
